# A vigilant observation to pregnancy associated listeriosis in Africa: Systematic review and meta-analysis

**Alene Geteneh**[1]*, **Sirak Biset**[2], **Selamyhun Tadesse**[1], **Alemale Admas**[3], **Abdu Seid**[4], **Demeke Mesfin Belay**[5]

**1** Department of Medical Laboratory Science, College of Health Sciences, Woldia University, Woldia, Ethiopia, **2** Department of Medical Microbiology, School of Biomedical and Laboratory Sciences, University of Gondar, Gondar, Ethiopia, **3** Department of Medical Laboratory Science, College of Medicine and Health Sciences, Bahir Dar University, Bahir Dar, Ethiopia, **4** Department of Midwifery, College of Health Sciences, Woldia University, Woldia, Ethiopia, **5** Department of Pediatrics and Child Health Nursing, College of Health Sciences, Debre Tabor University, Debre Tabor, Ethiopia

* aleneget@gmail.com

**Data Availability Statement:** All the necessary data are included in the paper.

**Funding:** The authors received no specific funding for this work.

## Abstract

The burden of human listeriosis, an emerging food-borne illness would be higher in Africa due to poor food processing practices. The severity of the disease and the high case fatality rate make human listeriosis an important public health problem. Besides, pregnant women and their fetuses are at higher risk of gaining human listeriosis. Thus, we planned to estimate the pooled prevalence of pregnancy-associated human listeriosis in Africa. Primary studies were exhaustively searched using PubMed, Cochrane, Web of Science, Google Scholar, and University of Gondar online research repository. Observational studies (cross-sectional) revealing the pregnancy-associated human listeriosis were incorporated. Eligible studies were selected and critically appraised for quality using the Joanna Briggs Institute (JBI) quality appraisal checklist. The required data were extracted and exported to Stata version 14 for meta-analysis. The pooled prevalence of pregnancy-associated human listeriosis in Africa was estimated using a weighted inverse random effect model. Sensitivity and sub-group analysis were conducted for evidence of heterogeneity. Among 639 reviewed articles, 5 articles were eligible with total study participants of 621. The pooled prevalence of pregnancy-associated listeriosis was found to be 5.17% (95% CI, 1.51, 8.82). The pooled level resistance of isolates was high. Cotrimoxazole and erythromycin were the relative choices of antibiotics for pregnancy-associated listeriosis in Africa. The burden of pregnancy-associated listeriosis in Africa was higher with an increased level of antibiotic resistance. Therefore, we recommend due attention to the deadly emerging disease in terms of health educations and the role of food hygiene particularly for risk groups, pregnant women. The antibiotics of choice should be after performing drug susceptibility test.

**Competing interests:** The authors have declared that no competing interests exist.

## Introduction

Human listeriosis is an important food-borne disease of public health problem responsible for high morbidity and mortality [1, 2]. Compared to the general population, pregnant women are twenty-folds at higher risk of developing listeriosis [2, 3], indicating that pregnancy by itself might be a risk factor for developing human listeriosis [4, 5]. The possible hematogenous dissemination of this maternal infection would end up with adverse pregnancy outcomes, such as miscarriage, stillbirths, and preterm labor [3, 6]. The recent report from China was the best example to show the hits of feto-neonatal mortality (57.1%) arose from perinatal listeriosis [7].

Because of the listeria's severe and fatal health consequences, particularly among infants, children, and elders, make listeriosis the most serious foodborne infections [8, 9]. More importantly, the US. National Institutes of Health (NIH), National Institutes of Allergy and Infectious Diseases (NIAID), and the Centers for Disease Control and Prevention (CDC) have been classified listeria monocytogenes, the main actor of human listeriosis, as the food and water-borne pathogens potential for bioterrorism threat [10–12].

The high incidence of pregnancy associated listeriosis (6.6% in Italy [13], 14.8% in India [14] and 16.9% in US), and the increased antibiotic resistance [15] remains the cause of public health threat. In Africa, the emerging food-borne illness like human listeriosis would be higher due to the poor food handling and processing practices as evidenced by the 2017–2018 South African outbreak [16]. The 2017–2018 South African outbreak indicated a case-fatality rate of 27%. Half of the cases (50%) were pregnancy-associated and 46% of pregnant girls and women have lost their fetuses [17]. Despite this concern, human listeriosis particularly pregnancy-associated listeriosis is not reported as such. This could be due to under-diagnosis and under-reporting of human listeriosis, the potential cause of miscarriage in human pregnancy [3, 18] or due to its mild or asymptomatic nature with no specific clinical symptoms [19]. Some of the studies presented an inconsistent magnitude of pregnancy-associated listeriosis: 1% in South Africa [20], 4.65% in West Africa [21], 5.56% in Ethiopia [22], 8.04% in Nigeria [23], and 8.51% in Ethiopia [24]. The prevalence revealed in these few and fragmented studies varied widely and remained indecisive. Thus, we planned to review articles in the area for precise estimates of the burden and better attention of the often-neglected human, pregnancy-associated listeriosis in Africa.

## Methods

### Reporting, protocol and registration

The protocol was designed based on Standard Preferred Reporting Items for Systematic Review and Meta-analysis (PRISMA) guide line [25] (S1 File). The protocol of this review was registered to the international prospective register of systematic reviews (PROSPERO, ID: CRD4 2021268855).

### Databases and search strategies

A comprehensive search using electronic databases such as, PubMed/ Medline, Cochrane, Web of Science, Google Scholar, and free Google access was performed. Besides, the University of Gondar online research repository were searched for unpublished sources. Reference lists and citations of included articles were used to find any other potentially significant articles. Content experts were communicated to search for additional studies, which were not salvaged by searching electronic databases and reference list searching. Conditions, Context, and Population (CoCoPop) search format was used. The included search term used a combination of relevant Medical Subject Headings (MeSH) and database specific terms with an Ethiopian

search filter. Search terms such as, "prevalence", "pregnant women", "listeria monocytogenes", "pregnancy-associated listeriosis" and "Africa" were used. Boolean operators, such as "AND" or "OR" were used to combine search terms. For insistence, the search strategy and the number of retrieved items for PubMed database has been presented (S2 File).

## Eligibility criteria

Observational studies (i.e., cross-sectional studies) reporting the prevalence of listeria monocytogenes among pregnant women in Africa were included. We included primary articles whose definition of listeriosis was pregnant women or newborns with L. monocytogenes in otherwise sterile body sites such as the blood, placenta or cerebrospinal fluid. Both mother and newborns are considered as a single case. For this review, published studies regardless of date of study using English language were included. However, those studies without full-text/abstract, non-English publications and studies not describing the outcome of the interest were not included. Case reports, editorials and qualitative reports were excluded.

## Outcomes of interest

The primary outcome of this study was pregnancy-associated listeriosis. The level of resistance for commonly tested antibiotics was the secondary outcome of interest for this review.

## Study selection and quality assessment

All retrieved studies were exported to EndNote version reference manager (Thomson Reuters, London) and duplicated studies were excluded. Two investigators (AG and AA) independently screened the titles and abstracts of full-text articles for eligibility. Two of the authors (AG and AA) independently evaluated eligible studies for quality using the Joanna Briggs Institute (JBI) quality appraisal checklist for prevalence studies [26]. Any variation that existed was resolved by a third author (D.M.B). Finally, studies which score are 50% and above quality assessment checklist criteria were considered as high quality.

## Data extraction

Two authors (AG and AA) extracted all the important data using standardized Microsoft Excel spreadsheet [27] after which cross-checking was done to maintain consistency. A third author (D.M.B) was involved as a tie breaker when discrepancies between the two authors were not brought to consensus through discussion. Data were extracted on first author name, year of publication, study area, prevalence of pregnancy-associated listeriosis with 95% CI, sample size, study design, types of specimen, and level of resistance for tested antibiotics. The encountered disparity and inconsistency among the authors were resolved by discussion and repeating the procedure. Whenever appropriate data were not available from the included studies, reviewers contacted the corresponding author and if no response was received the articles was excluded.

## Data analysis

The extracted data were exported to Stata version 14 statistical software for meta-analysis. Pooling effect sizes of listeriosis among pregnant women was computed using random-effect model due to substantial heterogeneity between the included primary studies ($I^2 = 77.9\%$, $p < 0.001$) [28]. Forest plot format was used to present the pooled estimate of pregnancy-associated listeriosis in Africa with 95% CI. $I^2$ statistic was calculated to assess the percentage of the total variation across studies (heterogeneity). The value of $I^2$ statistics; 25%, 50%, and 75%

represented low, moderate, and high heterogeneity respectively. P-value of $I^2$ statistic less than 0.05 was used to declare significant heterogeneity [28, 29]. The presence of publication bias was checked by observing the symmetry of the funnel plot [30]. Egger's test with a p-value less than 0.05 was also employed to determine if there was significant publication bias [31]. Sensitivity analysis was carried out since the value of $I^2$ statistic was greater than 50%, to identify the effect of individual studies on the pooled effect size of pregnancy associated listerosis. For evidence of substantial heterogeneity, sub-group analyses were performed using study setting and sample size as grouping variables. The findings are presented using figures, tables, and texts.

## Results

### Search results and characteristics of included studies

Six hundred thirty nine articles were retrieved from different searching electronic data base and University of Gondar online research repositories. After removing 49 duplicated articles, 590 articles were left. After screening 590 articles for titles and abstracts, 581 articles were obliterated. Nine articles were reviewed for full text and 4 articles were excluded based on the predetermined eligibility criteria. Finally, 5 articles were included for the analysis. The results of the search and reasons for exclusion during the study selection process is displayed at Fig 1.

A total of 621study participants were included. Of the included studies, two were from Ethiopia [22, 24] one each from South Africa [20], Nigeria [23], and West Africa [21]. All studies were cross-sectional study design with varying study periods from 1992 to 2021 (Table 1).

### Quality of the included studies

All the included studies were critically appraised using JBI quality assessment checklist adapted for cross-sectional type of study and scored for the validity of their results. Accordingly, among the five cross-sectional studies, two studies were scored six out of nine questions, 66.7% (low risk), one study was scored seven out of nine questions, 77.8% (low risk) and two studies were scored 9 out of nine questions, 100% (low risk). Therefore, from our quality appraisal, we generally found that all the included primary studies were of reliable in their methodological quality scores ranging between 6–9 from a total nine points for the cross-sectional studies. As a result, it is concluded that all the included studies had low risk of bias (high quality) (Table 2).

### Pooled prevalence of pregnancy-associated listeriosis

The pooled prevalence of pregnancy-associated listeriosis was 5.17% (95% CI, 1.51, 8. 82). The percentage of $I^2$ statistic indicates that the included studies are explained by significant heterogeneity ($I^2$ = 77.9, P < 0.001) Fig 2.

### Level of antibiotics resistance

The pooled level resistance for penicillin (66.7%), cotrimoxazole (41.6%), erythromycin (31.3%), clindamycin (52.1%) and ciprofloxacin (50%) were 46.3% Fig 3. While the individual studies displayed for higher rate of resistance(penicillin G (66.7%), clindamycin (66.7%), amoxicillin (50%) and vancomycin (50%) at Tigray, Ethiopia) [24] and 100% resistance to meropenem and penicillin G, ciprofloxacin (75%), cotrimoxazole/doxycycline (50%) erythromycin/clindamycin (37.5%), and gentamicin (25%) at Jimma, Ethiopia) [22]. Relatively gentamicin, erythromycin and cotrimoxazole were seen as the antibiotics of choice for pregnancy-associated listeriosis in Africa.

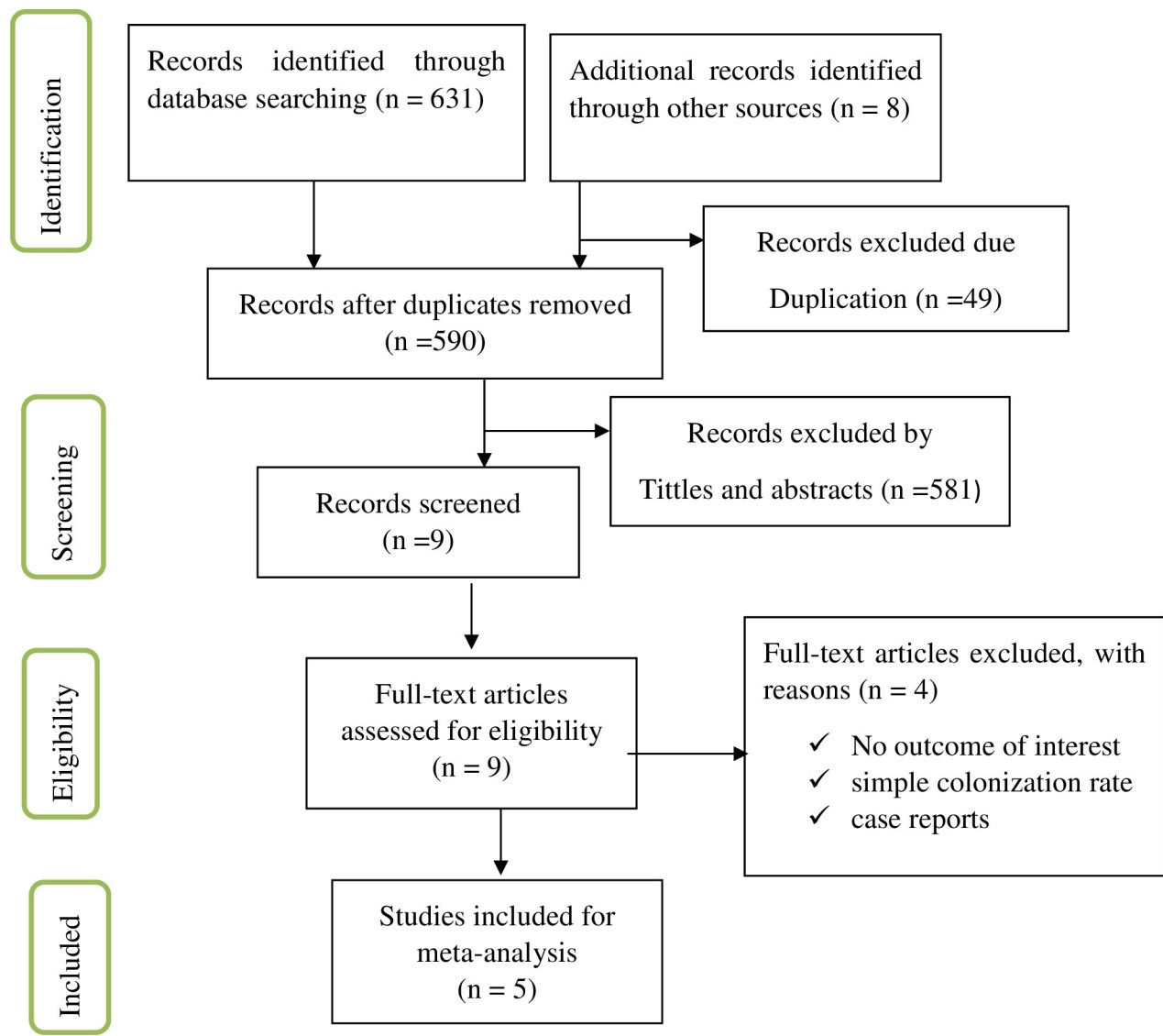

**Fig 1. PRISMA flow chart showing selection of articles for pregnancy-associated listeriosis in Africa, 2021.**

## Investigation of heterogeneity

The p-value and $I^2$ statistic of forest plot shows marked heterogeneity among the studies ($I^2$ = 77.9, P < 0.001). Hence, sub-group analysis by study country and sample size was performed to minimize heterogeneity (Table 3).

**Table 1. Characteristics of studies included in the systematic review and meta-analysis of pregnancy-associated listeriosis in Africa, 2021.**

| Author | Publication year | Country | Study design | Sample size | Quality score | Prevalence with 95% CI |
|---|---|---|---|---|---|---|
| Van rensburg [20] | 1992 | South Africa | Cross- sectional | 206 | 7 | 0.97 (0.9, 1.1) |
| Shindang et al. [23] | 2013 | Nigeria | Cross sectional | 87 | 6 | 8.04 (7.4, 8.7) |
| Welekidan et al. [24] | 2019 | Ethiopia | Cross-sectional | 141 | 9 | 8.5 (8.1, 8.9) |
| Fall et al. [21] | 2020 | West Africa | Cross sectional | 43 | 6 | 4.6 (3.6, 5.6) |
| Girma et al. [22] | 2021 | Ethiopia | Cross sectional | 144 | 9 | 5.56 (5.2, 5.9) |

**Table 2. Quality of included studies using JBI critical appraisal tool for studies reporting prevalence data.**

| Author (s) | Was the sample frame appropriate to address the target population? | Were study participants sampled in an appropriate way? | Was the sample size adequate? | Were the study subjects and the setting described in detail? | Was the data analysis conducted with sufficient coverage of the identified sample? | Were valid methods used for the identification of the condition? | Was the condition measured in a standard, reliable way for all participants? | Was there appropriate statistical analysis? | Was the response rate adequate, and if not, was the low response rate managed appropriately? | Score |
|---|---|---|---|---|---|---|---|---|---|---|
| Van Rensburg and Odendaal [20] | Yes | Yes | Yes | Yes | Yes | Yes | Yes | No | NA | 7 (77.8%) |
| Shindang et al. [23] | Yes | No | Yes | Yes | Yes | Yes | Yes | No | NA | 6 (66.7%) |
| Welekidan et al. [24] | Yes | Yes | Yes | Yes | Yes | Yes | Yes | Yes | Yes | 9 (100%) |
| Fall et al. [21] | Yes | Yes | NA | Yes | Yes | Yes | Yes | No | NA | 6 (66.7%) |
| Girma et al. [22] | Yes | Yes | Yes | Yes | Yes | Yes | Yes | Yes | Yes | 9 (100%) |

Note: NA = Not applicable

### Subgroup analysis by study country

According to country sub-group analysis, the highest pooled prevalence of pregnancy-associated listeriosis was found in Nigeria [8.05%: 95% CI; 2.33, 13.76] and the least was observed in South Africa [0.97%: 95% CI; -0.37, 2.31.7] (Table 3).

### Sub-group analysis by sample size

The pooled prevalence of pregnancy-associated listeriosis in studies conducted with sample size less than 100 was [6.51%: 95% CI; 2.28, 10.74], which is higher than the study conducted with sample size greater or equal to 100 [4.63%: 95% CI; -0.09, 9.35] (Table 3).

### Publication bias

Asymmetrical distribution of the funnel plot implies the presence of publication bias among the included studies Fig 4. Besides, the Egger's test result (p = 0.028) indicates significant small study effect bias when considering pregnancy-associated listeriosis studies (Table 4).

## Discussion

The annual report of 1,600 cases and 255 deaths in the United States [32], and the recent world's largest south African outbreak of 1060 cases and 216 deaths [33] indicated that listeriosis has emerged as the most common causes of morbidity and mortality globally. Besides, European case reports indicated that the incidence of human listeriosis increased from time to time particularly in Germany, France, and Finland from 1999 to 2003 [34]. This suggests the possibility of an increased burden of listeriosis in Africa, where poor food processing practices prevail.

This systematic review and meta-analysis publicized that the pooled estimate of pregnancy-associated listeriosis in Africa was 5.17% (95% CI: 1.51, 8.82). This burden is higher

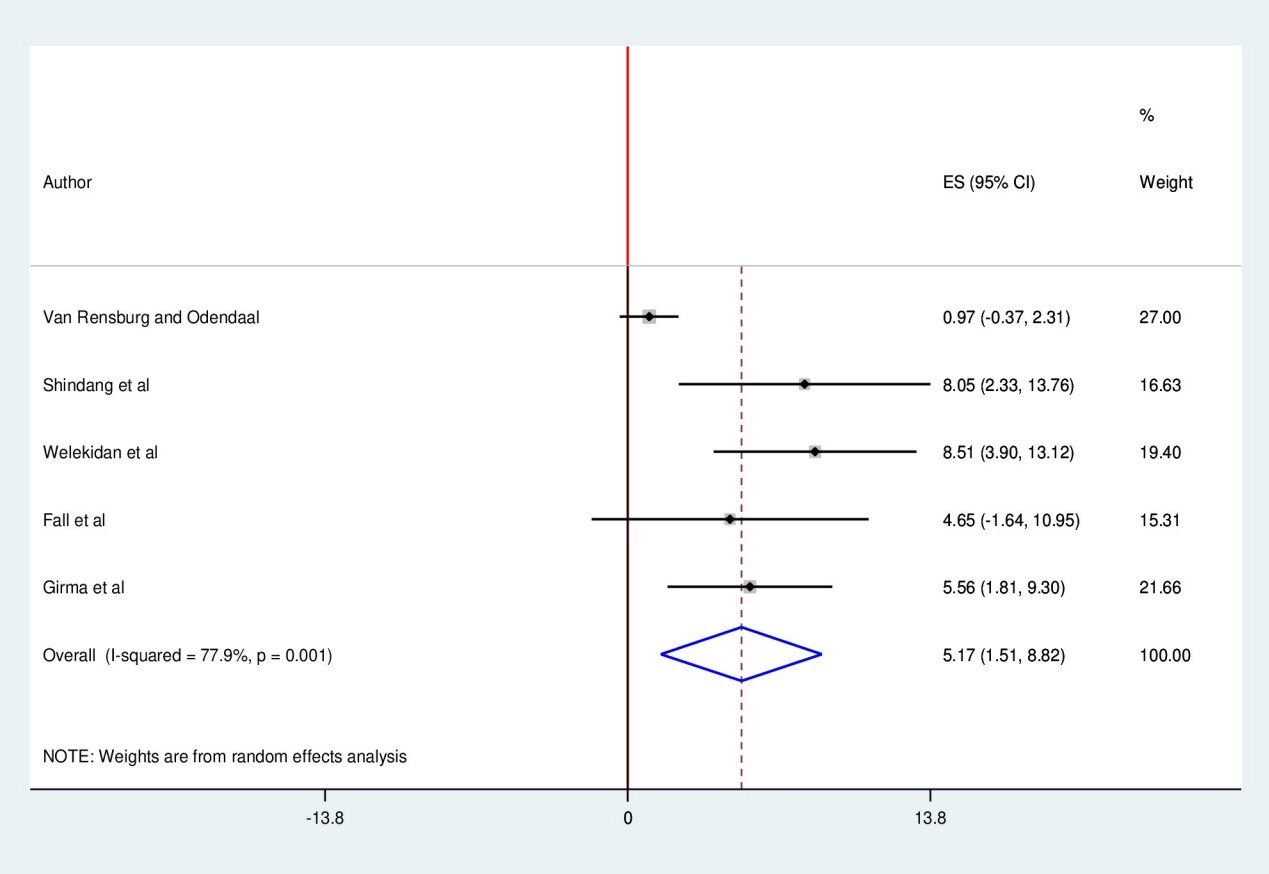

**Fig 2. Forest plot showing pooled prevalence of pregnancy-associated listeriosis in Africa, 2021.**

considering the severity and potential fatality of the disease. Thus, human listeriosis was attributing to more than 5% of pregnancy-related compilations in Africa including fetal loss [3, 16, 35]. This might be owing to the decreased function of cell mediated immunity during pregnancy and placenta, often immune privileged site preference of *Listeria monocytogenes* puts pregnant women and her fetus at risk of developing listeriosis [5, 36].

The human origin prevalence of listeriosis in Iran (5.5%) [37], 10% [38], and Indonesia (9%) [39] were consistent. The consistency of findings could brief the global emergence, and role of globalization in disease transmission. Our finding was lower compared to the annual 17% pregnancy-associated listeriosis in the USA [40]. The discrepancy might be due to exposure variability and the habit of eating ready-to-eat food stuffs.

More than 46% of pregnant girls and women have lost their fetuses by a single outbreak in South Africa [17]. Accordingly, one can estimate the fate of many pregnancy-associated listeriosis cases in this region. What fueled, the concern is that governments in Africa had no well-structured biodefense mechanisms, even not prioritized in their policy documents. The 5.2% pooled prevalence of pregnancy-associated listeriosis warrants the need for routine screening of suspected pregnant women, and incorporate in the mother to child priority diseases for early management and prevention. The under-diagnosis and under-reporting of human listeriosis, developing nation's health professional knowledge about pregnancy-related listeriosis and its health consequences to both the mother and her fetus, and asymptomatic infections in

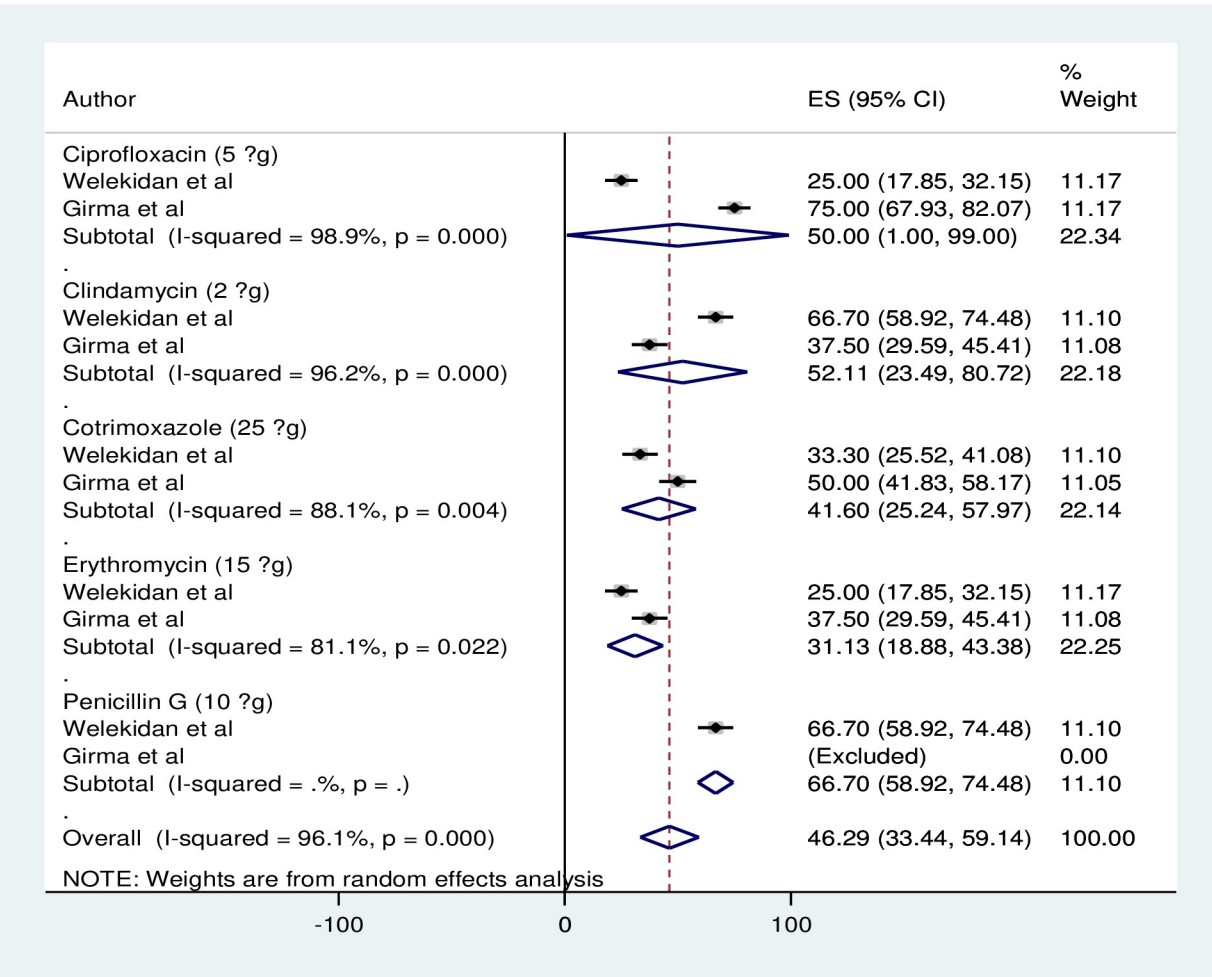

**Fig 3. Level of resistance for commonly tested antibiotics.**

Africa might mask the real deadly burden of pregnancy associated listeriosis in the region [3, 18, 19, 41]; which may also cause pregnancy-associated listeriosis to be remained neglected.

Antibiotics prescribed for pregnancy-associated listeriosis are selected based on their ability to penetrate into the host cell, bind the specific pencillin binding protein of listeria, and cross the placenta in adequate concentration [42]. Thus, penicillin G, ampicillin, amoxicillin, cotrimoxazole, or erythromycin, and some combined with gentamycin, can be used to treat

**Table 3. Subgroup analysis for a pooled estimate of pregnancy-associated listeriosis in Africa, 2021.**

| Variable | Characteristics | Study included | Sample size | Prevalence with 95% CI |
|---|---|---|---|---|
| Country | South Africa | 1 | 206 | 0.97 (-0.368, 2.31) |
| | Nigeria | 1 | 87 | 8.05 (2.33, 13.76) |
| | Ethiopia | 2 | 285 | 6.73 (3.83, 9.63) |
| | West Africa | 1 | 43 | 4.65 (-1.64, 10.95) |
| Sample size | >100 | 3 | 491 | 4.63 (-0.09, 9.35) |
| | <100 | 2 | 130 | 6.51 (2.28, 10.74) |
| Overall | | 5 | 621 | 5.17 (1.51, 8.82) |

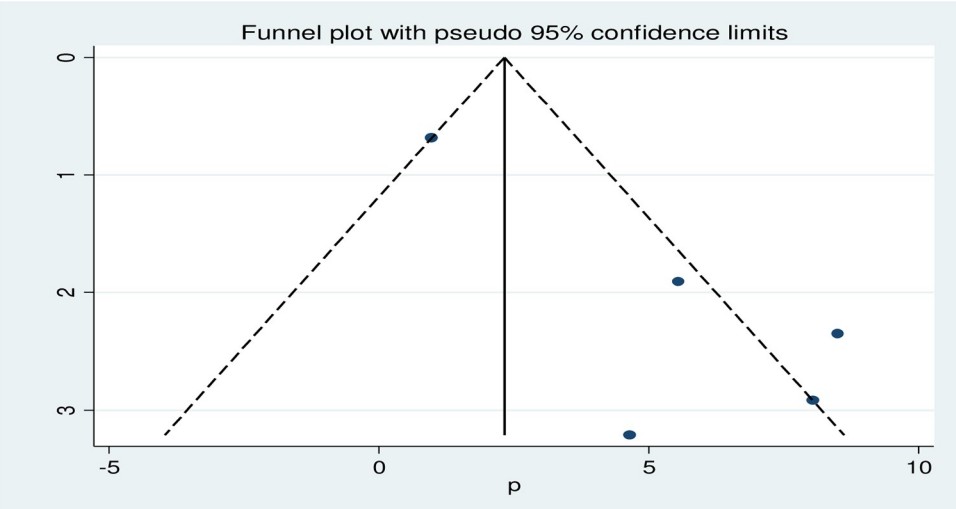

**Fig 4. Funnel plot to displaying the existence of publication bias.**

listeriosis during pregnancy [42, 43]. Treating listeriosis can also be difficult due to the bacterium's ability to produce biofilm [44, 45] and acquire antimicrobial resistance genes from nonlethal antimicrobial exposure or other pathogenic bacteria through horizontal gene transfer [46]. The pooled level resistance for commonly tested antibiotics was 46.3%, and the level of resistance was variable between individual studies Fig 3. Although individual studies recommended ciprofloxacin, erythromycin, cotrimoxazole and chlorampinicol [24], and ampicillin, gentamicin, rifampicin, erythromycin and clindamycin [22], the pooled resistance in this study indicated that cotrimoxazole and erythromycin were the relative choices of antibiotics for pregnancy-associated listeriosis in Africa. The variations in study period, population differences, sample size differences and years of publications might be the possible cause of heterogeneity among primary studies.

## Limitations of the study

This is the first review to assess the pooled prevalence of pregnancy associated listeriosis in Africa. All the primary studies included are hospital-based which affects representativeness to the community. Despite the potential of this study in generating African burden of pregnancy associated listeriosis, there were some shortcomings to be dealt with. Lack of primary studies from other African countries, and the smaller sample size of accessed studies were the main restrictions of this study. The application of random effect model, and the inability of the study to identify confounding factors might have an effect in presenting the true magnitude of pregnancy associated listeriosis. Therefore, we would like to direct our remembrance to the scientific community to be chary of deciphering the finding of this review considering the intrinsic limitations of original studies incorporated.

**Table 4. Eager test.**

| Std_Eff | Coef. | Std. Err. | t | P> |t| | [95% CI] |
|---|---|---|---|---|---|
| slope | -0.9854924 | 1.007895 | -0.98 | 0.400 | -4.193065, 2.22208 |
| bias | 3.036709 | 0.7559462 | 4.02 | 0.028 | 0.6309511, 5.442468 |

## Conclusion

The burden of pregnancy-associated listeriosis in Africa was higher with an increased level of antibiotic resistance. Therefore, we recommend due attention to the deadly emerging disease in terms of health educations and the role of food hygiene particularly for at-risk groups, pregnant women. The antibiotics of choice should be after performing drug susceptibility test.

## Supporting information

**S1 File. PRISMA guideline.**
(DOC)

**S2 File. Sample search strategy.**
(DOCX)

## Author Contributions

**Conceptualization:** Alene Geteneh.

**Formal analysis:** Alene Geteneh.

**Investigation:** Alene Geteneh.

**Methodology:** Alene Geteneh, Demeke Mesfin Belay.

**Software:** Demeke Mesfin Belay.

**Supervision:** Sirak Biset, Alemale Admas.

**Visualization:** Sirak Biset, Selamyhun Tadesse, Abdu Seid.

**Writing – original draft:** Alene Geteneh.

**Writing – review & editing:** Sirak Biset, Selamyhun Tadesse, Alemale Admas, Abdu Seid, Demeke Mesfin Belay.

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
