## [Decision Letter · Decision Letter 0]

8 Jul 2022

PGPH-D-22-00671

A Vigilant Observation to Pregnancy Associated Listeriosis in Africa: Systematic Review and Meta-Analysis

Dear Dr. %Alene Geteneh%,

Thank you for submitting your manuscript to PLOS Global Public Health. After careful consideration, we feel that it has merit but does not fully meet PLOS Global Public Health’s publication criteria as it currently stands. Therefore, we invite you to submit a revised version of the manuscript that addresses the points raised during the review process.

EDITOR: Please follow the reviewer's comment

We look forward to receiving your revised manuscript.

Kind regards,

M Abdullah Yusuf

Academic Editor

Journal Requirements:

1. Please amend your online Financial Disclosure statement. If you did not receive any funding for this study, please simply state: “The authors received no specific funding for this work.”

2. Please update your online Competing Interests statement. If you have no competing interests to declare, please state: “The authors have declared that no competing interests exist.”

3. Please provide separate figure files in .tif or .eps format and ensure that all files are under our size limit of 10MB.

4. We noticed that you used “unpublished sources/studies” in the manuscript. We do not allow these references, as the PLOS data access policy requires that all data be either published with the manuscript or made available in a publicly accessible database. Please amend the supplementary material to include the referenced data or remove the references.

5. We have noticed that you have uploaded Supporting Information files, but you have not included a list of legends. Please add a full list of legends for your Supporting Information files after the references list. 

We suggest you thoroughly copyedit your manuscript for language usage, spelling, and grammar. If you do not know anyone who can help you do this, you may wish to consider employing a professional scientific editing service.

Additional Editor Comments (if provided):

Please follow the comments and correct accordingly

Reviewers' comments:

Reviewer's Responses to Questions

**Comments to the Author**

1. Does this manuscript meet PLOS Global Public Health’s publication criteria? Is the manuscript technically sound, and do the data support the conclusions? The manuscript must describe methodologically and ethically rigorous research with conclusions that are appropriately drawn based on the data presented.

Reviewer #1: Yes

2. Has the statistical analysis been performed appropriately and rigorously?

Reviewer #1: Yes

3. Have the authors made all data underlying the findings in their manuscript fully available (please refer to the Data Availability Statement at the start of the manuscript PDF file)?

Reviewer #1: Yes

4. Is the manuscript presented in an intelligible fashion and written in standard English?

Reviewer #1: Yes

5. Review Comments to the Author

Reviewer #1: 1. The introduction should be more elaborated. The background of the study is missing which should be related to the burden of listeriosis among pregnant lady. But here the majority of the sentences are comparison which is not appropriate.

2. Why child are included in this study?

3. The section of eligibility criteria & outcome of interest are written in similar meanings. Please rewrite correctly.

4. The author has mentioned in the result section the time duration of 1988 to 2019 but in the table section, it is 1992 to 2021. explain why?

5. Description of antibiotic sensitivity pattern has been written in a very concise way which should be written elaborately.

6. PLOS authors have the option to publish the peer review history of their article (what does this mean?). If published, this will include your full peer review and any attached files.

**Do you want your identity to be public for this peer review?** For information about this choice, including consent withdrawal, please see our Privacy Policy.

Reviewer #1: **Yes: **Dr.Tarana Jahan

---

## [Decision Letter · Decision Letter 1]

8 Sep 2022

PGPH-D-22-00671R1

A Vigilant Observation to Pregnancy Associated Listeriosis in Africa: Systematic Review and Meta-Analysis

Dear Dr. Geteneh,

Thank you for submitting your manuscript to PLOS Global Public Health. After careful consideration, we feel that it has merit but does not fully meet PLOS Global Public Health’s publication criteria as it currently stands. Therefore, we invite you to submit a revised version of the manuscript that addresses the points raised during the review process.

Please find some final minor comments to respond to below. We look forward to seeing your revisions.

We look forward to receiving your revised manuscript.

Kind regards,

Julia Robinson

Executive Editor

Journal Requirements:

Additional Editor Comments (if provided):

Reviewers' comments:

Reviewer's Responses to Questions

**Comments to the Author**

1. If the authors have adequately addressed your comments raised in a previous round of review and you feel that this manuscript is now acceptable for publication, you may indicate that here to bypass the “Comments to the Author” section, enter your conflict of interest statement in the “Confidential to Editor” section, and submit your "Accept" recommendation.

Reviewer #1: All comments have been addressed

2. Does this manuscript meet PLOS Global Public Health’s publication criteria? Is the manuscript technically sound, and do the data support the conclusions? The manuscript must describe methodologically and ethically rigorous research with conclusions that are appropriately drawn based on the data presented.

Reviewer #1: Yes

3. Has the statistical analysis been performed appropriately and rigorously?

Reviewer #1: Yes

4. Have the authors made all data underlying the findings in their manuscript fully available (please refer to the Data Availability Statement at the start of the manuscript PDF file)?

Reviewer #1: Yes

5. Is the manuscript presented in an intelligible fashion and written in standard English?

Reviewer #1: Yes

6. Review Comments to the Author

Reviewer #1: The paper is nicely written.

There are some observation:

1. The introduction should be more elaborated. The background of the study is missing which should be related to the burden of listeriosis among pregnant lady. But here the majority of the sentences are comparison which is not appropriate.

2. Why child are included in this study?

3. The section of eligibility criteria & outcome of interest are written in similar meanings. Please rewrite correctly.

4. The author has mentioned in the result section the time duration of 1988 to 2019 but in the table section, it is 1992 to 2021. explain why?

5. Description of antibiotic sensitivity pattern has been written in a very concise way which should be written elaborately.

7. PLOS authors have the option to publish the peer review history of their article (what does this mean?). If published, this will include your full peer review and any attached files.

**Do you want your identity to be public for this peer review?** For information about this choice, including consent withdrawal, please see our Privacy Policy.

Reviewer #1: **Yes: **Dr. Tarana Jahan

---

## [Editor Report · Decision Letter 2]

23 Sep 2022

A Vigilant Observation to Pregnancy Associated Listeriosis in Africa: Systematic Review and Meta-Analysis

PGPH-D-22-00671R2

Dear Mr Geteneh,

We are pleased to inform you that your manuscript 'A Vigilant Observation to Pregnancy Associated Listeriosis in Africa: Systematic Review and Meta-Analysis' has been provisionally accepted for publication in PLOS Global Public Health.

Best regards,

Julia Robinson

Executive Editor